# Hypercholesterolemia in Cancer and in Anorexia Nervosa: A Hypothesis for a Crosstalk

**DOI:** 10.3390/ijms23137466

**Published:** 2022-07-05

**Authors:** Giulia Gizzi, Samuela Cataldi, Claudia Mazzeschi, Elisa Delvecchio, Maria Rachele Ceccarini, Michela Codini, Elisabetta Albi

**Affiliations:** 1Department of Philosophy, Social Sciences and Education, University of Perugia, 06126 Perugia, Italy; giulia.gizzi@studenti.unipg.it (G.G.); claudia.mazzeschi@unipg.it (C.M.); elisa.delvecchio@unipg.it (E.D.); 2Department of Pharmaceutical Sciences, University of Perugia, 06126 Perugia, Italy; samuela.cataldi@unipg.it (S.C.); mariarachele.ceccarini@unipg.it (M.R.C.); michela.codini@unipg.it (M.C.)

**Keywords:** cancer, hypercholesterolemia, anorexia nervosa, lipids, sphingomyelinase, keratinocytes

## Abstract

The relationship between cholesterol and cancer has been widely demonstrated. Clinical studies have shown changes in blood cholesterol levels in cancer patients. In parallel, basic research studies have shown that cholesterol is involved in the mechanisms of onset and progression of the disease. On the other hand, anorexic patients have high cholesterol levels and a high susceptibility to cancer. In this review, we first present a brief background on the relations among nutrition, eating disorders and cancer. Using several notable examples, we then illustrate the changes in cholesterol in cancer and in anorexia nervosa, providing evidence for their important relationship. Finally, we show a new possible link between cholesterol disorder in cancer and in anorexia nervosa.

## 1. Introduction

Appropriate nutrition is essential for health. Current scientific research focuses heavily on the nutrition–cancer relationship [1]. Eating disorders (EDs) already present during childhood, such as being overweight or underweight, were correlated with the onset of tumors [2]. The variability of eating habits in different countries could explain differences in tumor prevalence; a state of denutrition or malnutrition was present at the beginning of the disease [3]. In the pediatric population, EDs that interfere with the psychomotor, psychosocial and physical development of children, in addition with the development of cancer, have been described [4]. Karamanis et al. (2014) reported that cancer incidence was higher in patients with anorexia nervosa (AN) than in the general population [5]. Women with EDs had a high risk of breast cancer [6], although some studies showed a reverse association [7]. The early onset of AN was considered to be an important factor for the development of breast cancer [8]. Moreover, associations between EDs and the risks for esophageal cancer and stomach cancer were described [9].

Significant research has focused on lipids as key bioeffector molecules, resulting in the paradigm that they are involved in the pathogenetic mechanism for the onset and progression of cancer.

## 2. Cholesterol: An Intriguing Molecule

Cholesterol (Chol) represents one of the major eukaryotic lipids. Historically, Chol was first identified in the late 18th century by François Poulletier de la Salle. In the first part of the 19th century, Michel Eugène Chevreul created the name of “cholesterine” [10]. In subsequent years, this discovery was followed by elucidation of the importance of this very unique molecule for two specific reasons (Figure 1): (1) it plays a structural and functional role in itself; (2) it generates different types of molecules essential for cell health. In humans, Chol catabolism is limited, and this facilitates its pathological accumulation.

As a molecule in itself, Chol’s role depends on its structure that includes hydrophilic, hydrophobic and rigid domains. For this reason, Chol influences membrane fluidity and permeability [11] with dose-dependent effects [12]. Exciting aspects of Chol’s location in cell membranes include its ability to form lipid rafts [13], its role in membrane subcompartmentalization, functioning in endocytosis, signaling and many other membrane functions [14]. Additionally, the Chol identified in the inner nuclear membrane confers specificity to its action [15]. Here, it participates in the formation of nuclear lipid microdomains that act as platforms for active chromatin anchoring and for the regulated transcription process [16]. This particular role catapulted Chol into a class of bioactive lipids, which are types of molecules that are able to regulate gene transcription [17]. This property is used by Chol to regulate its own synthesis. The endoplasmic reticulum Chol level acts as a sensor for the intracellular Chol level. Thus, a reduced endoplasmic reticulum Chol concentration induces translocation of sterol regulatory element-binding protein transcription factor 2 from endoplasmic reticulum to Golgi apparatus, and then to the nucleus to activate the transcription of the gene encoding for 3-hydroxy-3-methylglutaryl-CoA reductase (HMGCR), a key enzyme for Chol synthesis [18]. In the plasma, Chol is transported by low-density lipoproteins (LDL), whereas it is removed from tissues and transported to the liver by high-density lipoproteins (HDL). Normal plasma levels of total Chol are less than 200 mg/dL; the 200–239 mg/dL range is considered borderline, while 240 mg/dL and above is considered high cholesterol [19]. In reality, hypercholesterolemia refers to the LDL value. It is considered a cardiovascular risk factor if (a) the LDL value is greater than 190 mg/dL in patients without other risk factors for the same pathologies as family history, hypertension, diabetes, smoking and low HDL levels; (b) the value is greater than 160 mg/dL in patients with one major risk factor, or greater than 130 mg/dL in patients with two cardiovascular risk factors [20].

Among the important findings is the demonstration that Chol is able to generate a multitude of functional molecules, such as oxysterols, steroid and sex hormones, vitamin D and bile acids [17].

## 3. Cholesterol Disorder in Cancer

It is becoming increasingly evident in the literature that dyslipidemia, especially Chol disorder, not only is a risk factor for cardiovascular disease, but is also a risk factor for the development and progression of cancer. In particular, dyslipidemia, as a comorbidity of obesity and diabetes, has been involved in breast cancer [21]. Interestingly, high levels of serum phospholipids, Chol, sphingolipids and eicosanoids in breast cancer patients, and their altered distribution in the alveolar spaces, were responsible for the increased risk in SAR-CoV-2 infection [22]. Moreover, dyslipidemia plays a role in prostate cancer [23], non-small cell lung cancer [24], in colorectal cancer [25] and in second cancers in thyroid cancer patients [26].

Genomic medicine studies have highlighted genetic variations in both familial hypercholesterolemia (FH) and cancer, such as Lynch syndrome and hereditary breast and ovarian cancer syndrome [27]. The use of statins in heterozygous FH did not induce significant reduction in the expected cancer mortality during the first period of treatment; however, in the second period a reduction of 37% was obtained [28]. However, Vaseghi et al., in a systematic review on familial FH comorbidities, reported that cancer had a lower or similar prevalence in FH patients with respect to the general population [29].

Many researchers have questioned the possible mechanism of Chol action in cancer. In a comprehensive review by Murai [30], papers reported increased Chol biosynthesis, and the role of Chol-enriched domains of the cell membrane in the regulation of transmembrane signaling, cell adhesion and migration. The role of Chol and sphingolipid-enriched lipid rafts was confirmed by studies that followed [31]. Moreover, Chol-enriched domains located in the inner nuclear membrane were found to play a role in cancer [32]. Additionally, 27-hydroxycholesterol was found to be an endogenous selective estrogen receptor modulator that may be relevant in endocrine cancer [33]. It was demonstrated that hypercholesterolemia induced an increase in Chol entry into cancer cells, since they use Chol for their vitality and proliferative activities [34]. The entry of Chol into cancer cells is important, as it is due to the overexpression of the LDL receptor (LDLR) in transformed cells [35]. As the disease progresses, Chol is heavily internalized, resulting in severe hypocholesterolemia. This blood parameter must never be underestimated, as it can be a marker for cancer progression [36,37]. Lymphoblastic lymphoma cells of the SUP-T1 cell line strongly increased DNA synthesis when conditions of hypercholesterolemia were reproduced in vitro [38]. At the same time, the Chol levels in the cells increased, while in the culture medium it reduced significantly [38]. In haematological malignancies, severe hypocholesterolemia was associated with hypophospholipemia and high levels of antiphospholipid antibodies [39].

Wang et al. [40] reported that the manipulation of lipid levels in patients with initial cancer by using pharmacological compounds may be a good strategy to combat this malignant disease. In fact, statin use has been associated with a reduced risk of cancer [41]. It is therefore important to monitor lipidemia, and particularly cholesterolemia, in subjects with a family history of cancer. In these patients and in patients with diabetes and dyslipidemia, the continued use of statins may reduce the risk of cancer [42]. Lee et al. [43] reported that a moderate or poor adherence to statin treatment increased the risk of both cardiovascular mortality and cancer. Moreover, the baseline serum Chol level is considered predictive of the clinical benefits for advanced non-small-cell lung cancer patients who underwent immune checkpoint inhibitor-based treatment [44]. In this regard, a relation between hypercholesterolemia and inflammation, activated by oxidated lipoproteins, was reported. In fact, the oxidation of LDL is considered to be responsible for the activation of many pathways associated with inflammation in breast cancer [45] and colorectal cancer [46]. Thus, hypercholestrolemia is able to induce a low-grade inflammatory state [47]. This condition facilitates the proliferation and the migration of tumor-associated macrophages and myeloid-derived suppressor cells to the tumor micro-environment [47]. The inhibition of Chol metabolism and its esterification in T-cells potentiates the antitumor response that CD8+ cells mediate [48]. Moreover, hypercholesterolemia was found to facilitate the response to cancer immunotherapy. Indeed, Perrone et al. reported the prognostic role of hypercholesterolemia in terms of overall survival and non-progression-free survival in patients treated with immune checkpoint inhibitors [49]. Recently, a research focus has been on proprotein convertase subtilisin/kexin-type 9 (PCSK9), a circulating negative regulator of LDL receptors that reduces blood Chol [50]. Genetic mutations responsible for amplification of PCSK9 activity are considered a potential cause of lethal familial hypercholesterolemia. Conversely, a reduction in PCSK9 activity induced an increase in hepatic LDL receptors, resulting in the lowering of blood Chol levels [51]. Interestingly, PCSK9 influences immune checkpoint regulation in cancer. Thus, PCSK9 monoclonal antibody or siRNA therapy that is currently used in clinics worldwide to treat hypercholesterolemia could be useful to study as a combined therapeutic strategy for cancer/metastasis [51].

## 4. Hypercholesterolemia in Anorexia Nervosa

AN is a severe psychiatric disease that is difficult to treat [52]. Symptoms include the refusal to maintain a physiologically healthy body weight, and an intense fear of becoming fat even if a subject is underweight [53]. In addition, patients manifest the disorder through self-induced vomiting, laxative abuse, a strict diet and intense/continuous physical activity. The non-recognition and misperception of one’s own body weight constitute the body dysmorphic disorder that is now part of the spectrum of ‘Obsessive Compulsive Disorder and Related Disorders’ within the Diagnostic and Statistical Manual of Mental Disorders (DSM-5) [54]. AN is a pathology that mainly affects girls and young women (0.5–1%) [55]. The age of onset is the beginning of adolescence, and the disorder’s severity crosses different levels, from a low severity to a high severity that leads to the death of the patient. AN is a disorder that is present to a greater extent in industrialized societies, where being thin is of considerable importance. The spread of the concept of thinness as the perfect constitution has been encouraged by the media and online content. Consequently, there has been an increase in eating disorders, even in societies where they used to be a rarity. The people most at risk are models, dancers and athletes (PDM-2) [56].

Many studies have used functional magnetic resonance imaging to investigate the neurobiological basis of AN. Until about 15 years ago, it was reported that the areas for the general perception of body image and the estimate of its size were localized in the extrastriate area, fusiform area and the lower parietal lobule of the brain, without significant differences between control and anorexic patients [57]. One of the reasons for this is that scientific research paid little attention to the different specific aspects of body image evaluation such as the perception of oneself and one’s body, and in the attitudes and emotions toward it [58]. In subsequent years, a very discussed symptom of AN was body image perception disturbance, a multidimensional model that included affective, cognitive, perceptual and behavioral components [59]. The cerebral cortical areas activated in the image perception disturbance are the following: a specific brain region located in the right lateral temporal-occipital cortex, the extrastriate body area [60,61] and a different area in the central part of the fusiform gyrus called the fusiform body area [62]. Moreover, the influence of the amygdala has also been demonstrated [63]. Attention has been paid to the emotional dimension of AN, showing that the frontal cortex, insula and prerirhinal cortex are brain regions that are activated in the body image evaluation process [64], with high differences observed between control and anorexic patients [65,66,67]. A study by Myers et al. [68] highlighted the activation of the frontal cortex and insula in anorexic patients whenever they look at a thin body. In the control group, the prerirhinal cortex is recruited whenever subjects see images of a fat body. The insula is therefore closely connected to emotions, and is probably more activated in anorexic patients than in controls due to a greater emotional value given to a lean body.

AN is characterized by changes in different biochemical–clinical parameters [69]. It is accompanied by secondary hyperlipoproteinemia, with increased synthesis of triglyceride-rich lipoproteins [70]. Interestingly, high levels of total Chol and LDL were observed [71,72]. Notably, the 4β-hydroxycholesterol/Chol ratio was higher in AN patients than in control subjects [73]. The hypercholesterolemia in AN patients was not due to a de novo synthesis [74], even if a secondary synthesis linked to the hyperglycemia consequent to the increased level in cortisol was possible [75]. Previously, Nestel [76] related hypercholesterolemia in AN to the diminished Chol and bile acid turnover as a compensatory mechanism to reduced caloric intake. Moreover, a high level of Chol ester transfer protein activity was demonstrated [77]. Interestingly, Chol levels were negatively correlated with suicidal behavior of patients [78]. The altered lipid profile in AN was associated with increased cardiovascular risk [79,80]. The development of inflammation and oxidative stress in AN has been described [81], but there are no data regarding its relation with hypercholesterolemia. The relation between hypercholesterolemia in AN and cancer has not been considered.

## 5. Hypercholesterolemia in Anorexia Nervosa and Susceptibility to Cancer: Hypothesis for a Crosstalk

At the apex of complexity is the direct demonstration that in AN patients the dislipidemia participates in the pathogenesis of cancer. As reported earlier, a large body of work has netted several significant advances in the study of cancer and anorexia nervosa showing that (1) dyslipidemia is involved in cancer; (2) patients with AN have dyslipidemia; (3) patients with AN are predisposed to some types of cancer. Our classical understanding of how lipid metabolites participate in signaling and cell regulation has been shaped to a large extent by the conceptualization that they are not only structural or energetic molecules, but are also functional molecules. This implies that the dysmetabolism in AN is not only responsible for an alteration of the energy supply, but also for a modification of the bioactive lipid molecules that may induce cell transformation. Inside the nucleus, Chol is partly free and partly linked to sphingomyelin [82]. During DNA synthesis, activation of sphingomyelinase degrades sphingomyelin and releases Chol. This process thus increases in relation to the S phase of the cell cycle [83]. Moreover, Chol accumulation leads EGFR/Src/Erk signaling, reactivation-mediated sphingosine-1-phosphate nuclear translocation, with consequent estrogen-related receptor alpha re-expression [84], a target for cancer therapy [85]. It is possible to hypothesize that the abnormal levels of Chol found in AN patients may be correlated with the nuclear processes of gene duplication and transcription. This could explain the increased susceptibility to specific tumors found in AN patients (Figure 2). At present, no studies on hypocholesterolemia in the advanced cancers of AN patients have been performed.

Furthermore, no data exist regarding the modulation of crosstalk hypercholesterolemia and cancer in AN patients undergoing therapy aimed at reducing blood levels of Chol. Future studies could be useful to study whether or not there is a reduced risk of cancer in AN patients undergoing drug therapies aimed at reducing hypercholesterolemia.

## 6. Concluding Remarks and Future Perspectives

Currently, a direct link between dyslipidemia in AN and susceptibility to cancer is missing. Understanding this connection can open an avenue to new therapies aimed at reducing cancer onset and its progression in AN patients. Firstly, a correct dietary intervention is essential to limit the increase in Chol found in patients with AN. Additionally, modulation of the lipid levels in AN patients is an exciting field of research and is potentially very feasible, given the ease with which it can be modulated. Currently, there are several strategies for doing this, including the use of products extracted from plants. Nutritional interventions and/or treatments with supplements aimed at regulating cholesterol levels could be strategies to consider, in order to reduce the increased susceptibility of cancer in patients with AN. Current knowledge requires the development of research in the field, with the development of new preventive/therapeutic approaches that are based on hypocholesterolemic products. A successful clinical translation of these new finding requires the initial study of a large population of patients with AN, in order for the results to become used as the gold standard practice throughout the world.

## Figures and Tables

**Figure 1 ijms-23-07466-f001:**
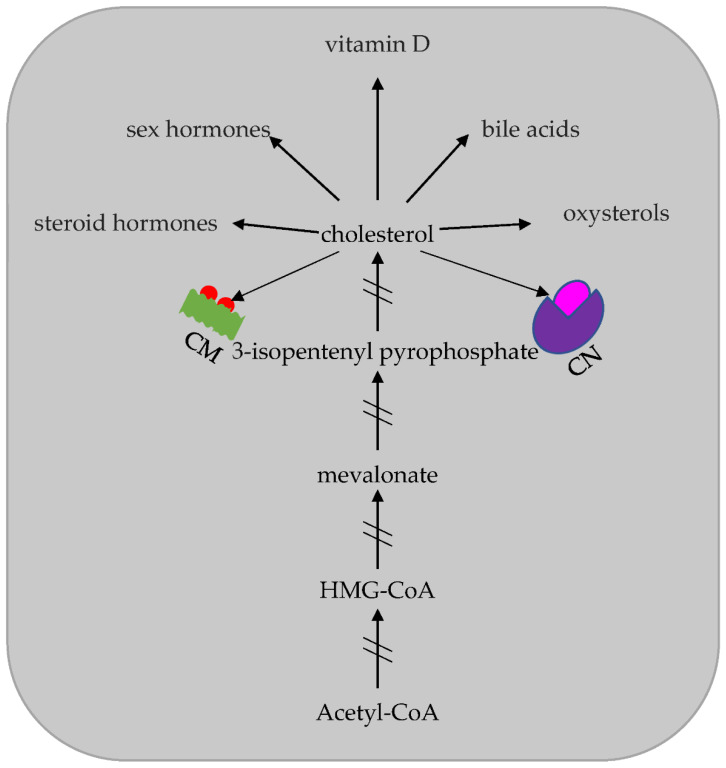
Schematic representation of the synthesis beginning from Acetyl-CoA, and the utilization of cholesterol.

**Figure 2 ijms-23-07466-f002:**
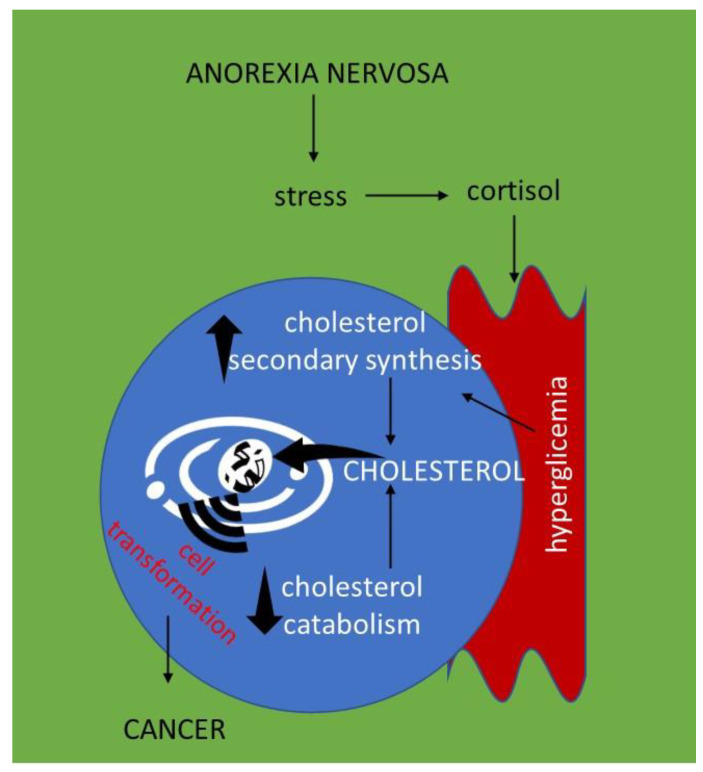
**Hypothesis of the relationship between hypercholesterolemia in anorexia nervosa and cancer susceptibility.** The increase in cholesterol in anorexia nervosa patients may be due to the following: (1) increased synthesis resulting from cortisol-induced hyperglycemia in response to stress [75]; (2) decreased catabolism [76]. The increase in cellular cholesterol content could be responsible for its nuclear transfer, where chromatin activity influences and induces cellular transformation that leads to the onset of cancer.

## Data Availability

Not applicable.

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
