# Peer review of "Hypercholesterolemia in Cancer and in Anorexia Nervosa: A Hypothesis for a Crosstalk"

_ijms, 2022, doi:10.3390/ijms23137466_

Round 1
Reviewer 1 Report
The submitted manuscript that aims to describe the potential link between hyperchol., AN and cancer is well written in general but still requires some details.
General comment:
- Section about cholesterol metabolism could be shortened as already very well described elsewhere
- Usefulness of section about hypercholes-induced hypochol in cancer questionable
- Authors should try to find evidence and discuss the prevalence of cancer in familial hypercholesterolemia. This would be a way to circumvent the potential role of suboptimal nutrition in cancer / and suboptimal nutritional status in AN.
- Authors should discuss link between hyperchol-induced inflammation and oxidative stress and cancer. What about inflammation and oxidative stress in AN?
- Figure 2 seems to consider increased de novo cholesterol synthesis and decreased bile acid turnover. However, lines 178-186 do not seem to put more importance on the latter.
Minor comments
- In the abstract, there seems to be missing a word in the sentence « …relation nutrition, eating disorders and cancer. »
- Cholesterol should not be abbreviated by CHO (rather used for carbohydrate).
- Line 28: Eds should be EDs
Author Response
Reviewer 1
Thanks for your comments that really improved the review
General comment:
- Section about cholesterol metabolism could be shortened as already very well described elsewhere
The section about cholesterol metabolism has been shortened and related references deleted (lines 104-109)
- Usefulness of section about hypercholes-induced hypochol in cancer questionable
Thanks for this remark. The specific paragraph has been removed. Some important studies on the hyper-hypocholesterolemia relationship in cancer have been merged into the previous paragraph (lines 159-160). Lack of studies on hypocholesterolemia in anorexic patients with advanced cancer has been reported (lines 299-300)
- Authors should try to find evidence and discuss the prevalence of cancer in familial hypercholesterolemia. This would be a way to circumvent the potential role of suboptimal nutrition in cancer / and suboptimal nutritional status in AN.
Your observation is very interesting for my studies. At the moment few data exist in the literature. These have been reported (lines 134-140)
- Authors should discuss link between hyperchol-induced inflammation and oxidative stress and cancer. What about inflammation and oxidative stress in AN?
The link between hypercholesterolemia-induced inflammation and oxidative stress and cancer has been reported (lines 170-189). No data exist about these aspects in AN ( 275-277)
- Figure 2 seems to consider increased de novo cholesterol synthesis and decreased bile acid turnover. However, lines 178-186 do not seem to put more importance on the latter.
It was our mistake. We thank the referee very much for this observation. Figure 2 was corrected
Minor comments
- In the abstract, there seems to be missing a word in the sentence « …relation nutrition, eating disorders and cancer. »
The statement has been corrected
- Cholesterol should not be abbreviated by CHO (rather used for carbohydrate).
Abbreviation has been substituted with Chol
- Line 28: Eds should be Eds
It has been corrected (line 46)
Reviewer 2 Report
In this interesting paper, the author summarized the possible crosstalk of hypercholesterolemia and cancer progression in anorexia nervosa. The paper is interesting, well performed and the english style is adequate. However, if present I suggest to the author to insert the study that described the role of lipid lowering therapies in this crosstalk. If not present, a future investigation of this topic could be considered in the Discussion.
Author Response
Reviewer 2
In this interesting paper, the author summarized the possible crosstalk of hypercholesterolemia and cancer progression in anorexia nervosa. The paper is interesting, well performed and the english style is adequate. However, if present I suggest to the author to insert the study that described the role of lipid lowering therapies in this crosstalk. If not present, a future investigation of this topic could be considered in the Discussion.
Thank you very much for this observation. At the moment do not exist data in the field. It has been included in the discussion (lines 301-304)